# The Impact of Vp-Porin, an Outer Membrane Protein, on the Biological Characteristics and Virulence of Vibrio Parahaemolyticus

**DOI:** 10.3390/biology13070485

**Published:** 2024-06-28

**Authors:** Jinyuan Che, Qitong Fang, Shaojie Hu, Binghong Liu, Lei Wang, Xiu Fang, Lekang Li, Tuyan Luo, Baolong Bao

**Affiliations:** 1Key Laboratory of Yangtze River Water Environment, Ministry of Education, College of Environmental Science and Engineering, Tongji University, Shanghai 200092, China; 18616829916@163.com (J.C.); celwang@tongji.edu.cn (L.W.); 2Key Laboratory of Exploration and Utilization of Aquatic Genetic Resources, Ministry of Education, National Demonstration Center for Experimental Fisheries Science Education, Shanghai Ocean University, Shanghai 201306, China; 19853565191@163.com (Q.F.); 15233813696@163.com (S.H.); m230100053@st.shou.edu.cn (B.L.); 3Fujian Provincial Key Laboratory of Breeding Lateolabrax Japonicus, Fuding 355200, China; minwei@minwei.cn; 4Jiujiang Academy of Fishery Sciences, Jiujiang 332000, China; lekangli1987@126.com; 5Institute of Quality Standards and Testing Technology for Agro-Products, Fujian Academy of Agricultural Science, Fuzhou 350003, China

**Keywords:** *Vibrio parahaemolyticus*, Vp-porin, antimicrobial resistance, motility, virulence

## Abstract

**Simple Summary:**

The increasing antibiotic resistance of *Vibrio parahaemolyticus* (*V. parahaemolyticus*), a critical halophilic pathogen, has become a significant concern, highlighting the need for a deeper understanding of the molecular mechanisms governing bacterial drug resistance in *V. parahaemolyticus*. Porins, essential proteins located in the outer membrane, play a direct role in influencing antimicrobial resistance mechanisms in bacteria. This study aimed to characterize a novel porin and evaluate its role in antimicrobial resistance by constructing a deletion mutant (∆*Vp-porin*). The results showed that ∆*Vp-porin* exhibited impaired membrane integrity and increased susceptibility to certain antibiotics. Furthermore, the motility of ∆*Vp-porin* was impaired, and its virulence was attenuated, as assessed by *Tetrahymena*. These findings demonstrate the significant role of Vp-porin in modulating antimicrobial resistance and bacterial virulence.

**Abstract:**

Porins are crucial proteins located in the outer membrane that directly influence antimicrobial resistance mechanisms and virulence in bacteria. In this study, a porin gene (*Vp-porin*) was cloned in *V. parahaemolyticus*, and the function of Vp-Porin in biological characteristics and virulence was investigated. The results of sequence analysis showed that Vp-Porin is highly conserved in *Vibrio* spp., and the predicted 3D structure showed it could form a 20-strand transmembrane β-barrel domian. Membrane permeabilization provides evidence that the membrane integrity of ∆*Vp-porin* was damaged and the sensitivity to tetracycline, polymyxin B, rifampicin and cephalothin of ∆*Vp-porin* obviously increased. In addition, loss of *Vp-porin* damaged motility due to downregulated flagellar synthesis. In addition, ∆*Vp-porin* exhibited attenuated cytotoxicity to Tetrahymena. The relative survival rate of Tetrahymena infection with *∆Vp-porin* was 86%, which is much higher than that with WT (49%). Taken together, the results of this study indicate that Vp-Porin in *V. parahaemolyticus* plays various roles in biological characteristics in membrane integrity, antimicrobial resistance and motility and contributes to virulence.

## 1. Introduction

Vibrio parahaemolyticus is a Gram-negative bacterium commonly associated with gastroenteritis due to the consumption of contaminated seafood [1]. Sepsis resulting from gastroenteritis and wound infections can be fatal, especially in individuals with underlying liver conditions. Traditionally, antibiotics have been the primary treatment for *V. parahaemolyticus* infections [2]. However, studies have shown that the widespread multi-drug resistance observed in *V. parahaemolyticus* strains isolated from both environmental and clinical sources is largely attributed to the inappropriate use of antibiotics in aquaculture production [3,4,5,6].

The emergence of antibiotic resistance in *V. parahaemolyticus* poses a significant public health challenge, highlighting the necessity for a thorough understanding of the molecular mechanisms driving bacterial drug resistance. *V. parahaemolyticus* exhibits notable environmental adaptability, closely linked to its effective and precise regulatory system [7,8]. The evolution of antibiotic resistance in bacteria is a multifaceted process encompassing diverse mechanisms, including porin-mediated efflux pumps, target modification, enzymatic degradation or alteration, cell wall and membrane adjustments and horizontal gene transfer [9,10,11]. Among these mechanisms, the interaction between porins and antimicrobial compounds plays a significant role in the development of resistance [7]. However, there is limited research on the molecular mechanism of antibiotic resistance in *V. parahaemolyticus*; understanding these mechanisms is essential for effective management and control of infections.

Porins are outer membrane proteins in Gram-negative bacteria that play a crucial role in modulating cellular permeability and antibiotic resistance. These transmembrane, pore-forming proteins with a β-barrel structure form water-filled channels for the passive transport of hydrophilic compounds. Porins can be classified into non-specific or specific types based on their activity and as monomeric, dimeric or trimeric based on their structural arrangement [9,12,13,14]. The function of porins in antibiotic resistance among Gram-negative bacteria lies in their role in mediating the entry of antibiotics across the outer membrane. Mutants lacking specific porins, such as OmpF, have been associated with antibiotic resistance in pathogens like *Escherichia coli*, Serratia marcescens, *V. parahaemolyticus* and Enterobacter aerogenes [15,16,17,18]. Conversely, the deletion of porins like OmpU has been shown to increase susceptibility to antibiotics, as observed in V. cholerae [19]. Despite the known impact of porins on antibiotic resistance in various bacteria, the understanding of their role in *V. parahaemolyticus* remains limited. Specifically, the contribution of specific porins to antibiotic resistance in this bacterium is not well elucidated. Additionally, porins located on the bacterial outer membrane can have implications beyond antibiotic resistance, influencing factors like flagellum function, virulence, protein export and adhesion, thus affecting the bacterium’s adaptability to external conditions [19,20,21,22].

In this study, we aimed to identify a novel porin gene in *V. parahaemolyticus* and explore its role in antibiotic resistance and other characteristics of the bacterium. By constructing a *Vp-porin* gene deletion mutant using overlapping PCR and two-step homologous recombination, we found that the deletion mutant displayed altered susceptibility to different antibiotics. Moreover, the mutants showed regulatory effects on flagellar synthesis and virulence in *V. parahaemolyticus*. These findings contribute to a deeper understanding of the role of porins in *V. parahaemolyticus* and their impact on bacterial adaptation and antibiotic response.

## 2. Materials and Methods

### 2.1. Strains, Media and Experimental Animals

The strains and plasmids utilized in this investigation are detailed in Table 1. For the construction of deletion mutants and subsequent functional analyses, we employed the *Vibrio parahaemolyticus* strain ATCC^®^ 17802™ (Guangdong Culture Collection Centre of Microbiology, Guangzhou, China). *V. parahaemolyticus* and its derived mutants were cultured in Luria–Bertani (LB) medium supplemented with 3% NaCl, incubated at 37 °C with continuous shaking at 150 rpm. *Tetrahymena thermophila*, provided by Prof. Shan Gao from the Ocean University of China, was used for comparative studies. Tetrahymena was axenically cultured in an SPP medium at a temperature of 30 °C.

### 2.2. Protein Domain and Structure Analysis

The amino acids of Vp-porin were determined from the genome of ATCC (ID:17802) and validated by PCR and sequencing. The ID of Vp-Porin protein is AMG08901.1. To investigate the conservation of protein sequences across different genera, we employed Clustal Omega X2.1 and ESPript 3.0 for generating a multiple sequence alignment of deduced protein sequences of Vp-Porin, which were retrieved from the NCBI database for various Vibrio spp. For the construction of the 3D structure of Vp-Porin, homology modeling was carried out using the Swiss Model server (http://swissmodel.expasy.org), with the protein 3D model template obtained from the Protein Data Bank (PDB) server (https://www.wwpdb.org).

### 2.3. Construction of ∆Vp-porin Deletion Mutant and Complement Strain

The *Vp-porin* gene sequence was obtained from the genome of the ATCC17802 strain and validated by PCR and sequencing (the sequence is listed in the Appendix A). The gene ID of *Vp-porin* is NP_800037.1. Then, the *Vp-porin* gene was deleted using an allelic replacement strategy. The primers designed for constructing the *Vp-porin* mutant strains are listed in Table 2. PCR amplification was performed to obtain upstream and downstream flanking fragments of the target gene, generating 772 bp upstream and 917 bp downstream overlap fragments through overlap PCR. In detail, the two fragments for fusion should be amplified before the preparation of the overlap extension PCR (using primers up-F/up-R and down-F/down-R). For the overlap extension PCR, the upstream and downstream fragments were first added to the tube and subjected to 8 cycles of PCR reaction without primers for two-fragment fusion (up-R and down-F are reverse complementary). Then, primers up-F and down-R were added to the tube starting from the 9th cycle, and the PCR reaction was continued for an additional 30 cycles. These overlapped DNA products from *Vp-porin* were digested with *Sac* I and *Spe* I enzymes and subsequently inserted into the pSR47s plasmid at the *Sac* I and *Spe* I restriction sites. The resulting recombinant plasmid, pSR47S-∆*Vp-porin*, was transformed into the CC118 λpir strain and validated through gene sequencing. Following validation, the pSR47S-∆*Vp-porin* recombinant plasmid was introduced into the wild-type (WT) strain via conjugation and selected on LB agar plates containing kanamycin (Kan) and ampicillin (Amp). The second cross-over recombination event was then selected for on LB agar containing 10% sucrose to isolate the ∆*Vp-porin* mutant through sucrose resistance screening. The mutation in the ∆*Vp-porin* strain was confirmed by PCR using primers T1/T2 and subsequent sequencing. The complementation strain was constructed according to the method of Zhang et al. (2012). A PCR fragment was amplified with the primers UP-F and Down-R, using DNA from the WT strain that contained the complement sequence of the *Vp-porin* gene. The fragment was ligated to pSR47S after excision with *Sac* I and *Spe* I. The complementation plasmid pSR47S-*Vp-porin-C* was transformed into the ∆*Vp-porin* via conjugation. The complement strain was checked by PCR using the primers UP-F and Down-R.

To assess the growth kinetics of the different strains, overnight bacterial cultures were diluted at a 1:100 ratio into 15 mL of fresh LB medium supplemented with 3% NaCl. The cultures were incubated at 37 °C with shaking at 150 rpm until they reached an OD600 value of 1.0. Subsequently, these cultures were further diluted 1:100 into 100 mL of the same medium and grown in a temperature-controlled incubator. Samples were collected every hour to monitor growth.

### 2.4. Proteolysis Activity Assay

The assessment of protease production and enzymatic activity was performed using LB agar plates supplemented with 2% skim milk. Overnight cultures strains were adjusted to an optical density at 600 nm (OD600) of 2.0. Following this adjustment, 5 μL aliquots of these diluted cultures were spotted onto separate agar plates and incubated at 37 °C for a duration of 6 h. The presence of proteolytic activity was determined by observing clear zones around the bacterial colonies. The diameters of these transparent zones were quantitatively measured using ImageJ software (ImageJ v1.47, NIH, Stapleton, NY, USA).

### 2.5. Outer Membrane Permeabilization Assay

The outer membrane permeability of bacteria was assessed using the N-phenyl-1-napthylamine (NPN) uptake assay, following established procedures [23]. Initially, cells cultured overnight were washed and suspended in buffer (5 mM HEPES, 5 mM glucose, pH 7.4). Subsequently, NPN was introduced to 2 mL of cells in a quartz cuvette to achieve a final concentration of 10 mM, and the background fluorescence was recorded (excitation wavelength = 350 nm, emission wavelength = 420 nm). Changes in fluorescence were monitored using BioTek Synergy HTX (Agilent, Beijing, China).

### 2.6. Morphological Observation

Bacterial cell morphology was evaluated using a scanning electron microscope (SEM) and transmission electron microscope (TEM). For SEM examination, the dehydrated samples were treated twice with 100% tert-butanol and dried with a freeze dryer for 4 h. Then, samples were placed on stubs and coated with gold film by sputter coating and viewed using a field emission scanning electron microscope (SEM, Hitachi SU8010, Tokyo, Japan). For TEM examination, all bacteria were inoculated onto 1% agar plates and incubated for 12 h at 37 °C. Samples were negatively stained with 1% uranyl acetate and visualized by electron microscope (TEM, Tecnai G2F30, FEI Co., Hillsboro, OR, USA).

### 2.7. Antimicrobial Susceptibility Testing and Survival Assay

The antimicrobial resistance of the WT and mutant to chloramphenicol, tetracycline, streptomycin, kanamycin, polymyxin B, rifampicin, cephalothin, levofloxacin, ampicillin and amoxicillin was determined by Antimicrobial Susceptibility Testing (AST). The experiment was carried out using the Kirby–Bauer Disk Diffusion Susceptibility Test method [24]. Two strains were cultured overnight and adjusted to an OD600 nm of 1. Then, the bacteria strains were spread on the LB agar (Merck, Darmstadt, Germany) using a sterile cotton swab. Sterile antimicrobial susceptibility disks were then placed on the agar plate and incubated at 37 °C for 12 h. The zone of inhibition was observed after 12 h of incubation. All assays were performed in triplicate. For survival assays, overnight cultures were adjusted to an OD600 nm of 0.5 in LB and diluted at a ratio of 1:100. Aliquots of 1 mL of culture were added to different antibiotics. Titrations of tetracycline were performed using concentrations ranging from 0.15 to 1.25 μg/mL, and concentrations ranging from 0.3 to 2.5 μg/mL were used for rifamycin. Cultures were incubated overnight at 37 °C in a rotary shaker, after which the colony-forming units were calculated by plating serial dilutions on LB agar. The percentage of bacterial survival was calculated by comparing the CFU/mL of the treated versus the untreated.

### 2.8. Motility Assay

Overnight cultures of WT, ∆*Vp-porin* and ∆*Vp-porin-C* strains were first adjusted to an optical density at 600 nm (OD600) of 1.0. Subsequently, 2 μL aliquots of these diluted cultures were spotted onto separate plates for motility assays. For assessing swimming motility, an LB medium containing 0.3% agar was used, while an LB medium with 1.5% agar was employed for evaluating swarming motility. The plates were then incubated at 37 °C for 12 h. Following the incubation period, the swarming capabilities of the strains were observed and recorded. To ensure the accuracy and reproducibility of the results, each experiment was conducted five times.

### 2.9. qRT-PCR Analysis

Total RNA extraction was performed utilizing the RNeasy Plus Mini Kit (Qiagen, Hilden, Germany), and genomic DNA contamination was eliminated using RNase-free DNase I. For cDNA synthesis, equal quantities of RNA (1 μg) from each sample were reverse transcribed using the PrimeScript RT Reagent Kit with gDNA Eraser (Takara, Tokyo, Japan). The specific primers employed for the qRT-PCR are detailed in Table 3. The qRT-PCR reactions were executed using the SLAN 96S Real-Time PCR System (Xiamen Zeesan Biotech, Xiamen, China) and FastStart Universal SYBR Green Master (Yeasen, Shanghai, China). To quantify the relative expression levels of the target genes, the transcript levels were normalized to those of the 16S rRNA gene, and the 2^−ΔΔCt^ method was applied for this calculation. To ensure the robustness and reproducibility of the results, three independent biological replicates were used for each experiment, with each individual replicate run in triplicate.

### 2.10. Assessment of Strains’ Virulence Using Tetrahymena thermophila

The virulence of the ∆*Vp-porin* mutant was assessed via the Tetrahymena infection model by the relative survival of bacteria and Tetrahymena after co-culturing as previously described [25]. Briefly, Tetrahymena was cultured in a sterile SPP medium at 30 °C for 48 h using an initial inoculum of 1 × 10^3^ cells/mL. Cells in this culture were harvested by centrifugation at 2000× *g* for 10 min at 10 °C, washed twice with sterile SPP medium and adjusted to 1 × 10^5^ cells/mL. After being cultured overnight, each strain was harvested, washed twice in SPP medium and adjusted to 3 × 10^9^ CFU/mL. Then, 5000:1 co-cultures of *V. parahaemolyticus* (3 × 10^9^ CFU/mL) and Tetrahymena (1 × 10^5^ cells/mL) were mixed and cultured 6 h at 30 °C. Bacterial growth in these mixed cell suspensions was determined every 1 h by measuring the absorbance at 450 nm every 1 h. Controls contained the bacterial strains mixed with an equal volume of SPP medium. Sterile SPP medium was used as the blank well. The relative survival of bacteria (%) was counted as the number of bacteria remaining in the culture relative to the number of bacterial cells grown alone. The relative survival of Tetrahymena (%) was calculated as the number of *Tetrahymena* cells remaining in the culture relative to the number of cells cultured alone. The cellular morphology of *Tetrahymena* cells incubated for 6 h was examined under a light microscope (Nikon 80i, Nikon Co, Tokyo, Japan) after they had been fixed with 4% PFA. Each experiment was repeated at least three times.

### 2.11. Statistical Analysis

Statistical analysis was performed by GraphPad Prism 9 (Graph Pad Software, Inc., San Diego, CA, USA). Experimental data were expressed as the mean ± SD. Two-group comparisons were analyzed using one-way ANOVA. ns; *p* > 0.05; * *p* < 0.05; ** *p* < 0.01; *** *p* < 0.001.

## 3. Results

### 3.1. Vp-porin Sequence and Structural Analysis

The *Vp-porin* gene from *V. parahaemolyticus* encompasses an open reading frame (ORF) spanning 1026 bp, encoding a protein of 342 amino acids with an estimated molecular weight of 37 kDa. A comprehensive multiple sequence alignment of the amino acid sequences revealed significant homology between Vp-Porin and porins from other Vibrio species, including *V. antiquarius* (82.95%), *V. harveyi* (86.30%), *V. owensii* (88.89%), *V. jasicida* (87.83%), *V. vulnificus* (64.53%), *V. campbellii* (85.47%) and *V. alginolyticus* (82.95%) (Figure 1A). The secondary structural elements of Vp-Porin, aligned above the sequence comparison in Figure 1A, suggest it may comprise three α-helices and twenty extended strands. The three-dimensional (3D) structure of Vp-Porin (Figure 1B,C) reveals a hollow β-barrel channel comprising 16 long β-strands and 4 short strands. The four short strands are shown in Figure 1C using black arrows. The depiction of the monomer emphasizes Vp-Porin’s role as a transmembrane channel, underscoring its significance in facilitating the transport of molecules across the bacterial membrane. In a word, Vp-Porin is a highly conserved protein that is speculated to function as an outer membrane porin, playing an important role in the regulation of cellular permeability and potentially impacting antibiotic resistance mechanisms.

### 3.2. Construction and Characterization of the Deletion Mutant of Vp-porin Gene and Complement Strain

To investigate the function of the *Vp-porin* gene, the *Vp-porin* deletion mutant of *V. parahaemolyticus* was constructed using an allelic replacement strategy. The primers utilized for the construction of the *Vp-porin* mutant are detailed in Table 2. Following the successful deletion of *Vp-porin*, the nucleotide fragment size of *Vp-porin* was reduced from its original 1029 bp to 78 bp (Figure 2A). The complementation strain construction method was the transformation of *V. parahaemolyticus* with a complementation construct directly integrating the *Vp-porin* gene into a chromosome. PCR verification confirmed the successful creation of the mutant strain ∆*Vp-porin* and its complementation strain (∆*Vp-porin-C*) (Figure 2B). As demonstrated by SEM, the WT, ∆*Vp-porin* and ∆*Vp-porin-C* strains displayed intact cell membranes and smooth surfaces, with no observable difference (Figure 2C). Nonetheless, the growth curve analysis revealed that the mutant exhibited a marginally slower growth rate compared to the WT strain and ∆*Vp-porin-C* strain, with OD600 values of 0.913 and 0.931 for the WT and ∆*Vp-porin* strains, respectively, after 12 h of cultivation at 37 °C (Figure 2D). Further analysis focused on the protease activity of ∆*Vp-porin* in the presence of 2% skim milk. The ∆*Vp-porin* strain displayed almost no transparent zone after incubation for 6 h. In contrast, wild-type and complementation strains exhibited distinct transparent zones (Figure 2E,F). This indicates a proteolytic deficiency in the ∆*Vp-porin* strain on a skim milk plate. These findings underscore that Vp-Porin may play an important role in facilitating essential physiological functions within *V. parahaemolyticus*.

### 3.3. Permeabilization of Outer Membranes

The outer membrane permeabilization of WT and mutant was determined by using the NPN uptake assay. NPN is a nonpolar hydrophobic fluorescent probe that is typically blocked by the outer membrane. However, when the integrity of the cell membrane is disrupted, it can enter the outer membrane and exhibit higher fluorescence intensity. Compared to WT and ∆*Vp-porin-C*, the fluorescence intensity of ∆*Vp-porin* was significantly higher (Figure 3). The results indicated that the *Vp-porin* gene is essential for the outer membrane permeability of *V. parahaemolyticus*.

### 3.4. Vp-porin Gene Deletion Affects Antimicrobial Susceptibility of V. parahaemolyticus

To determine whether the *Vp-porin* gene deletion mutant exhibited altered outer membrane properties, the susceptibility of WT, ∆*Vp-porin* and ∆*Vp-porin-C* strains to antibiotics was analyzed. As shown in Figure 4A,B, ∆*Vp-porin* showed significantly increased sensitivity to tetracycline, streptomycin, polymyxin B, rifampicin and cephalothin, and it displayed slightly increased sensitivity to chloramphenicol. However, there was no change in sensitivity to kanamycin, levofloxacin, ampicillin and amoxicillin. Survival curves of WT and ∆*Vp-porin* in the presence of varying concentrations of rifampicin and tetracycline were tested (Figure 4C). Upon exposure to 0.125 μg/mL rifampicin, WT and ∆*Vp-porin-C* exhibited survival rates of 55% and 38%, respectively, whereas the ∆*Vp-porin* mutant exhibited a survival rate of 17%. This indicated that the ∆*Vp-porin* mutant displayed a lower survival rate than the WT and ∆*Vp-porin-C* strains when exposed to rifampicin. After exposure to 0.625 μg/mL tetracycline, WT and ∆*Vp-porin-C* had a 35% and 27% survival rate, respectively, whereas the ∆*Vp-porin* mutant exhibited a low survival rate of 7%. In addition to that, we also confirmed the mRNA expression level of the two genes adjacent to *Vp-porin* using qRT-PCR analysis. The results indicated the mRNA expression levels of both *ompC* and *rhaT* genes were downregulated in *∆Vp-porin* (the result is supplied in Appendix A). This suggests the ∆*Vp-porin* mutant displays increased susceptibility to rifampicin and tetracycline. Overall, our results revealed that Vp-porin contributes to antimicrobial resistance.

### 3.5. The Vp-porin Mutant Exhibits Lower Motility and Decreases Transcription of Polar Flagellar Genes and Lateral Flagellar Genes in V. parahaemolyticus

To evaluate the swimming and swarming capabilities, we compared the flagellar production and motility of ∆*Vp-porin* with the WT and ∆*Vp-porin-C* strains. The WT and ∆*Vp-porin-C* strains displayed initial inoculum spreading throughout the plate in an asymmetrical pattern, characteristic of swimming (Figure 5A). Conversely, the ∆*Vp-porin* strain exhibited minimal movement beyond the initial inoculum site, indicating a defect in swimming in 0.3% agar plates. Similarly, when colonies of the WT, ∆*Vp-porin-C* and ∆*Vp-porin* strains were inoculated onto 1.5% agar plates and incubated, the WT and ∆*Vp-porin-C* strains exhibited colony growth expanding in a symmetrical pattern, indicating swarming behavior. In contrast, the colonies of the ∆*Vp-porin* strain exhibited limited movement on the plate, suggesting a slight defect in swarming (Figure 5A). Statistical analysis of the zone of swimming and swarming further confirmed that the ability of ∆*Vp-porin* to swim and swarm was defective (Figure 5B). Next, we used negative-staining electron microscopy to visualize the flagella of WT ∆*Vp-porin-C* and ∆*Vp-porin* strains. Whereas WT and ∆*Vp-porin-C* cells showed multiple flagella and produced lateral flagella, ∆*Vp-porin* produced fewer flagella and did not produce lateral flagella (Figure 5C). *V. parahaemolyticus* harbors both polar flagella and lateral flagella. We assessed the mRNA expression level of flagellar-related genes by qRT-PCR analysis. Notably, all polar flagellar cluster I genes (*flgB*, *flgM*, *flgK*, *flgC*) exhibited significant downregulation in the ∆*Vp-porin* strain compared with the WT strain, while polar flagellar cluster II genes *fliE* and *fliK* showed no significant difference between WT and ∆*Vp-porin* (Figure 5D). Furthermore, qRT-PCR was employed to assess the regulation of Vp-porin on lateral flagellar gene clusters. The results demonstrated significant downregulation of *flgA*, *flgB*, *flgG*, *lafA*, *motY* and *fliE* genes in the ∆*Vp-porin* strain compared to the WT (Figure 5E). Additionally, qRT-PCR analysis was also used to examine genes from the T6SS2, which plays a crucial role in virulence and adhesion. The mRNA expression levels of all genes (VPA1043, VPA1044, VPA1045) and *rpoN* were markedly downregulated in the ∆*Vp-porin* strain compared to the WT (Figure 5F). This indicated that Vp-Porin plays an important role in regulating the expression of T6SS2 genes. These results suggest that the motility defect caused by mistranslation is due to impaired flagellar assembly.

### 3.6. Assessment of Virulence of ∆Vp-porin Using Tetrahymena

To investigate the virulence of the ∆*Vp-porin* strain, a co-culture experiment was conducted using *Tetrahymena thermophila*. In the control group without bacteria, *Tetrahymena* cells exhibited an elliptical or pear-shaped morphology, with a large number of cells displaying rapid activity (Figure 6A). However, when co-cultured with bacteria, *Tetrahymena* cells underwent morphological changes, becoming shrunken and round, with increased vacuoles in the cytoplasm compared to the control group. Some cells even showed signs of cell death. Notably, co-culturing *Tetrahymena* with the WT strain resulted in severe cell shrinkage, deformation, and significant inhibition of cell growth. In contrast, when co-cultured with the ∆*Vp-porin* strain, *Tetrahymena* exhibited relatively improved growth, with decreased abnormal cell morphology and increased cell numbers observed compared to the WT group (Figure 6A). These observations suggest that the ∆*Vp-porin* strain possesses reduced virulence compared to the WT strain, as evidenced by the ameliorated effects on *Tetrahymena* growth and morphology. During infection, both the WT and the ∆*Vp-porin* mutant cultivated alone exhibited slow growth in a sterile SPP medium. However, the biomass of both the WT and the ∆*Vp-porin* mutant decreased continuously when co-cultured with *Tetrahymena*, suggesting that a large number of bacteria were preyed upon by *Tetrahymena* (Figure 6B,C). Notably, the biomass of the ∆*Vp-porin* strain declined more rapidly than that of the WT strain when co-cultivated with the same number of *Tetrahymena* (Figure 6C), indicating that the ∆*Vp-porin* strain was less resistant to predation by *Tetrahymena* compared to the WT strain. As depicted in Figure 5D, the relative survival of the ∆*Vp-porin* strain significantly decreased in the co-culture model, highlighting its reduced resistance to predation by *Tetrahymena* in comparison to the WT strain. The growth dynamics of *Tetrahymena* were also investigated in the co-culture model (Figure 6E). When cultured alone, the biomass of *Tetrahymena* increased during incubation, reaching a maximum concentration of 3.96 × 10^5^ cells/mL at 6 h. However, the growth of *Tetrahymena* was inhibited when co-cultivated with both the WT and ∆*Vp-porin* strains (Figure 6E). The number of *Tetrahymena* cells grown in the presence of WT initially increased before declining, with a relative survival rate of 49% at 6 h. In contrast, the number of *Tetrahymena* cells co-cultured with the ∆*Vp-porin* strain continued to increase throughout the incubation period. Co-culture with the ∆*Vp-porin* strain had a lesser impact on *Tetrahymena* viability, as evidenced by a relative survival rate of 86% (Figure 6F). In total, these data indicate the virulence of ∆*Vp-porin* is greatly attenuated.

## 4. Discussion

In this study, we aimed to explore the influence of porin on *Vibrio parahaemolyticus* by generating a ∆*Vp-porin* mutant. Our results uncovered that the absence of Vp-Porin led to increased susceptibility to some antibiotics and hindered both swimming and swarming capabilities. Additionally, utilizing a *Tetrahymena* infection model, we demonstrated that the deletion of *Vp-porin* significantly diminishes the virulence of *V. parahaemolyticus*.

The *Vp-porin* gene in *V. parahaemolyticus* displays significant sequence homology to porins found in various Vibrio species, suggesting a conserved structural and functional role for Vp-Porin across these organisms. Its predicted secondary structure and 3D configuration, characterized by 20 antiparallel strands forming a hollow β-barrel with distinct periplasmic and extracellular features, indicate potential channel-forming characteristics. The increasing prevalence of antimicrobial resistance (AMR) among bacterial pathogens, especially Gram-negative bacteria, emphasizes the necessity for a comprehensive understanding of the mechanisms contributing to this phenomenon [18,21,26]. Porins represent major proteins present in the outer membrane and play a direct role in antimicrobial resistance mechanisms [9]. The distinct role of outer membrane porins in antibiotic resistance and membrane integrity in *Escherichia coli* has been confirmed. Choi et al. put forward that porins can be classified into three groups according to their roles in antibiotic transport and membrane integrity: antibiotic-transport-related specific porins, membrane-integrity-related non-specific porins, and non-specific porins involved in both antibiotic transport and membrane integrity [14]. In this study, the sensitivity of the ∆*Vp-porin* strain to tetracycline, polymyxin B, rifampicin and cephalothin was significantly increased, revealing that Vp-Porin is involved in regulating antimicrobial resistance in *V. parahaemolyticus*, which is contrary to mutants of OmpF porins that the increased resistance impact to several antibiotics [14]. Antimicrobials can penetrate the outer membrane by two different pathways, through the lipid bilayer or through porins.

Research showed mutants of OmpU porin and OmpA contribute to increased susceptibility to some antibiotics due to impaired membrane integrity, which is similar to Vp-Porin in *V. parahaemolyticus* [14,19]. In the case of membrane-integrity-related non-specific porins, impaired membrane integrity can increase the intracellular diffusion of antibiotics [23]. An increase in the NPN fluorescence of ∆*Vp-porin* indicated the membrane integrity may be disrupted. Thus Vp-Porin is likely to be a membrane-integrity-related non-specific porin. Interestingly, no difference in sensitivity to kanamycin, ampicillin and amoxicillin was observed between the WT and ∆*Vp-porin* strains, indicating a specific role of Vp-Porin in modulating resistance to certain classes of antibiotics. Generally, chemicals with a molecular weight of more than 600 Da cannot penetrate the envelope of Gram-negative bacteria [14]. But the molecular weights of kanamycin, ampicillin and amoxicillin are 484, 349 and 365 Da, respectively, which are much less than 600 Da. We speculate there are other resistance mechanisms existing in *V. parahaemolyticus*, such as the presence of drug-resistance genes.

Furthermore, we demonstrated that the *Vp-porin* deletion mutant has few flagella and defective swimming motility, suggesting a potential role of Vp-Porin in flagellar synthesis and motility regulation. This phenomenon was also reported in ompX and ompA, which are critical for flagellar assembly and swimming ability in *Stenotrophomonas maltophilia* and *E. coli* [19,21,22]. Swimming motility is a critical aspect of bacterial pathogenesis. *V. parahaemolyticus* possesses a dual flagellar system, a polar flagellum for swimming in liquids and peritrichous lateral flagella for swarming over surfaces or in viscous liquids [27]. T6SS2, a major virulence determinant in *V. parahaemolyticus*, plays a role in bacterial invasion into host cells [28,29]. Our qRT-PCR analysis exhibited that the expression of T6SS2 genes was downregulated in the ∆*Vp-porin* strain compared to the WT strain, suggesting Vp-Porin could positively regulate the expression of T6SS2. It has been confirmed the downregulation of the *rpoN* gene is a critical factor responsible for the defective swimming motility of Δ*ompA* mutant [22]. We tested the mRNA expression level of *rpoN* and showed it was downregulated in the ∆*Vp-porin* strain. In addition, polar flagellar genes in cluster I and lateral flagellar genes in the ∆*Vp-porin* strain were also significantly downregulated. The sigma factor RpoN plays a vital role in regulating motility. The deletion of *rpoN* rendered *V. parahaemolyticus* nonmotile and downregulated polar flagellar systems and lateral flagellar systems [30]. These results indicate the defective motility of ∆*Vp-porin* might be correlated with *rpoN.* Precisely how *Vp-porin* deletion downregulates rpoN expression remains unclear. Above all, our results further suggest a multifaceted role of Vp-Porin in modulating gene expression profiles crucial for flagellar assembly, motility and virulence in *V. parahaemolyticus*.

*Tetrahymena*, a single-celled ciliate, has emerged as a valuable model organism for studying host–pathogen interactions. Recent research has utilized *Tetrahymena* to investigate bacterial infection mechanisms and host defense responses, shedding light on fundamental aspects of microbial virulence [31]. In this study, we aimed to evaluate the virulence of ∆*Vp-porin* using a *Tetrahymena* infection model. Similar to *Tetrahymena*–Aeromonas co-culture models [25], significant differences were observed in the relative survival of *Tetrahymena* co-cultured with the WT strain compared to ∆*Vp-porin*. Compared with the WT group, which exhibited shrinkage, deformation and a lower cell count, *Tetrahymena* co-cultured with ∆*Vp-porin* showed slightly improved growth and decreased abnormal cell morphology. This phenomenon is reminiscent of experiments involving Listeria monocytogenes, where hemolytic *L. monocytogenes* induces lysis of *Tetrahymena* pyriformis, while only a few protozoa undergo lysis in the presence of nonhemolytic *Listeria innocua* [32].

*Tetrahymena* prey on microorganisms and use phagocytosis to ingest and degrade these microorganisms [33]. In the process of predating, *Tetrahymena* use their oral groove to ingest food particles, forming initial food vacuoles. These initial food vacuoles move within the cell and fuse with digestive enzyme vesicles, forming mature food vacuoles where digestion occurs to produce nutrients [34,35]. However, the efficacy of this process can be affected by the nature of the bacteria consumed by *Tetrahymena*. It is likely that bacterial pathogenic mechanisms have been developed to resist predation by these predators [32]. Generally, microbial virulence factors encompass a wide range of molecules produced by pathogenic microorganisms, enhancing their ability to evade their host defenses and cause disease. Bacteria produce various virulence factors, such as hemolysin, the type III secretion system, the type VI secretion system, adhesion factors, iron uptake systems, lipopolysaccharides, proteases and outer membrane proteins [36]. Bacterial flagella and rpoN are important factors regulating bacterial adhesion, with flagella also being significant virulence factors [30]. In this study, we found a significant decrease in bacterial flagella synthesis and extracellular protease activity in *∆Vp-porin*, which may be related to the reduced virulence of *∆Vp-porin* towards *Tetrahymena*.

Previous molecular studies and animal infection experiments have indicated the critical role of certain porins, such as OmpX and OmpA from *E. coli*, and Acinetobacter baumannii, in bacterial pathogenicity by influencing bacterial adhesion and virulence factors [21,22,37]. Furthermore, it has been observed that OmpU from toxigenic strains evolves in the environment and serves as a preadaptation to virulence in the context of the human host [38]. Similarly, we found the ∆*Vp-porin* strain exhibits attenuated virulence compared to the WT strain, indicating Vp-Porin plays an important role in the virulence of *V. parahaemolyticus*. *V. parahaemolyticus* has many virulence factors; T6SS2 and adhesion factors are crucial factors controlling virulence. T6SS2 is also a mediator of *V. parahaemolyticus* adhesion to host cells [29]. In addition, the flagellum also affects bacterial virulence by promoting early biofilm formation and promoting adherence and invasion in *V. parahaemolyticus* [2,39]. Our results confirm that Vp-Porin can downregulate the expression level of T6SS2 genes and flagellar synthesis genes. Thus, the virulence of ∆*Vp-porin* strain is highly likely to be associated with the downregulation of T6SS2 and blocked flagellar synthesis.

## 5. Conclusions

In conclusion, we constructed a *Vp-porin* deletion mutant and preliminarily investigated the effects of the *Vp-porin* gene on antimicrobial resistance and virulence-associated properties in *V. parahaemolyticus*. The present results suggest that Vp-Porin modulates antimicrobial resistance and positively regulates flagellar synthesis in *V. parahaemolyticus*. Further analysis using a Vibrio–*Tetrahymena* co-culture model demonstrated that Vp-Porin could contribute to the virulence of *V. parahaemolyticus*. These findings not only are helpful for better understanding the function of Vp-Porin, but also provide further potential evidence supporting the feasibility of engineering strategies aimed at mitigating the antimicrobial resistance of *V. parahaemolyticus*.

## Figures and Tables

**Figure 1 biology-13-00485-f001:**
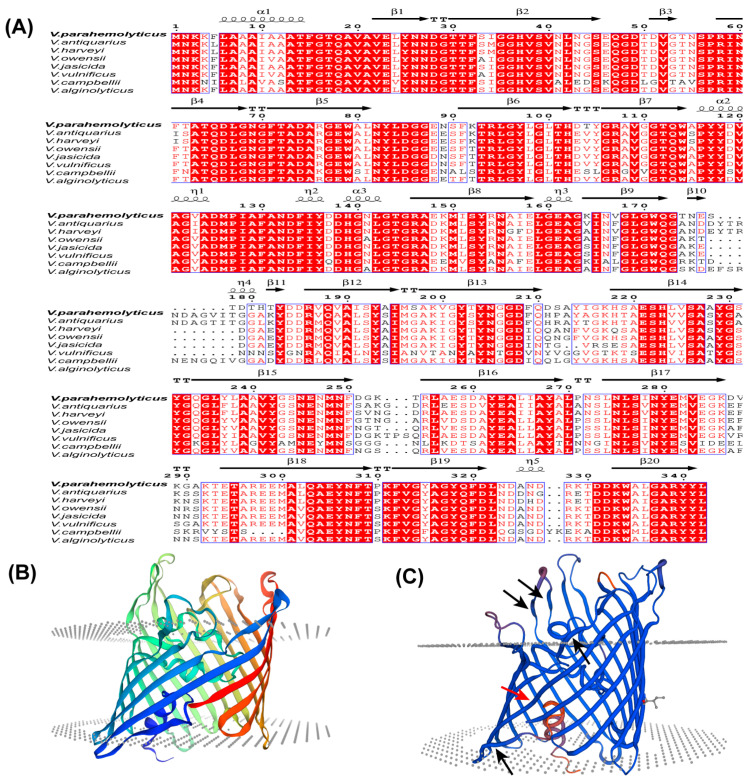
Homology modeling of the *V. parahaemolyticus* Vp-Porin structure. (**A**) Multiple sequence alignment of Vp-Porin in different vibrio species. Amino acid conservation is shown in red, and the secondary structure is based on the *V. parahaemolyticus* Vp-Porin structure. The η symbol refers to a 310-helix. Helices and strands are shown as black helices with squiggles and arrows, respectively. “.” indicates that there is a gap in the sequence compared to other sequence; (**B**,**C**) Three-dimensional structure of Vp-Porin using Swiss-model. Red arrow represents long β-strand. Black arrows represent short β-strands.

**Figure 2 biology-13-00485-f002:**
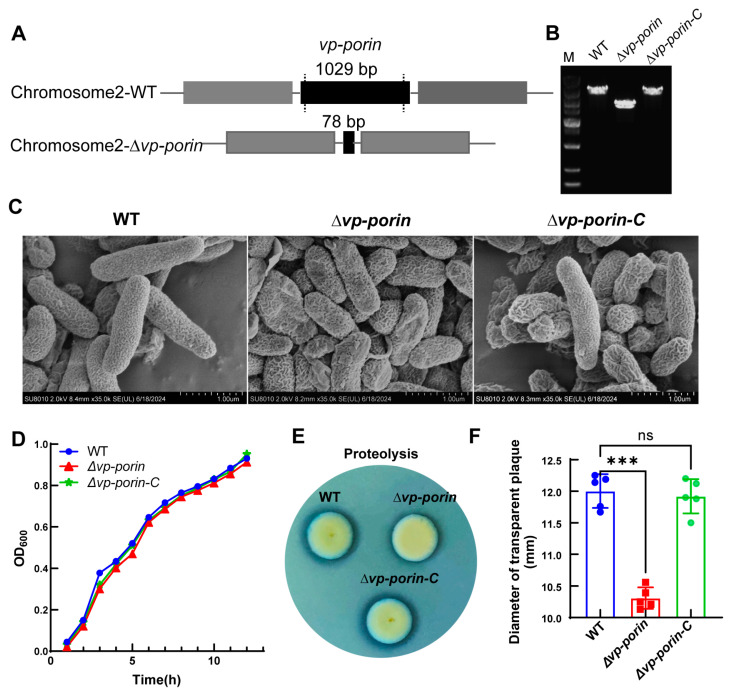
Construction of the deletion mutant strain of *V. parahaemolyticus* and phenotype characterization of ∆*Vp-porin*. (**A**) Construction of strain of *V. parahaemolyticus*. (**B**) The deletion of *Vp-porin* was confirmed by colony PCR. (**C**) SEM images of the WT, ∆*Vp-porin* and ∆*Vp-porin-C*; scale bar = 1 μm. (**D**) Growth curves in 3% NaCl LB medium over 12 h period. (**E**) Protease production on 2% skim milk agar. (**F**) Zone of proteolysis (mm) surrounding bacterial colonies after 6 h incubation at 37 °C. Columns have been marked with an asterisk (*** *p* < 0.001).

**Figure 3 biology-13-00485-f003:**
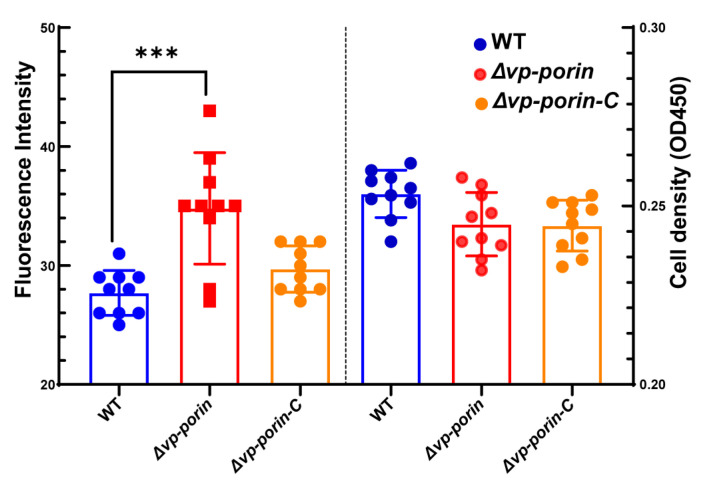
Comparison of outer membrane permeability of WT, ∆*Vp-porin* and ∆*Vp-porin-C*. The fluorescence intensity was determined with excitation at 350 nm and emission at 428 nm. Each bar represents the mean ± SD. *n* = 10. Columns have been marked with an asterisk (*** *p* < 0.001).

**Figure 4 biology-13-00485-f004:**
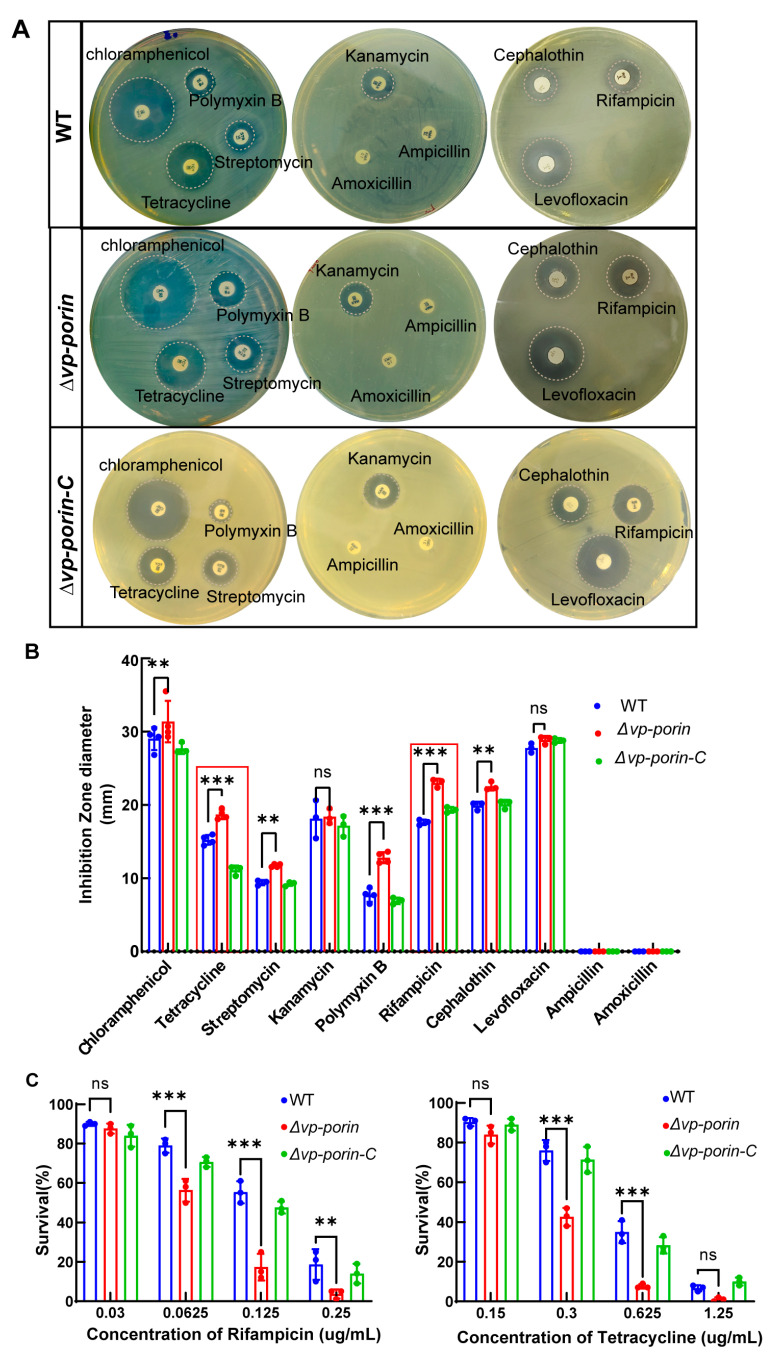
Antimicrobial activity of WT, ∆*Vp-porin* and ∆*Vp-porin-C*. (**A**) Antimicrobial screening showing zones of inhibition; (**B**) inhibition zone was calculated to determine the antimicrobial activity of WT, ∆*Vp-porin* and ∆*Vp-porin-C* towards various antimicrobials; (**C**) survival of WT, ∆*Vp-porin* and ∆*Vp-porin-C* in the presence of varying concentrations of rifampicin and tetracycline. Columns have been marked with an asterisk (** *p* < 0.01; *** *p* < 0.001).

**Figure 5 biology-13-00485-f005:**
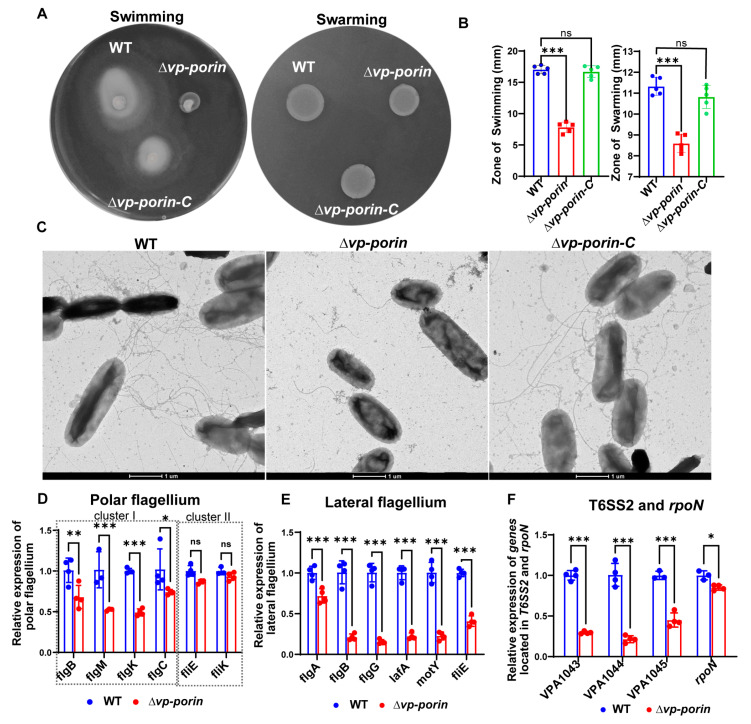
Vp-Porin regulates swimming and swarming motility in *V. parahaemolyticus*. (**A**) Swimming motility assay and swarming motility assay of WT, ∆*Vp-porin* and ∆*Vp-porin-C* on LB plates with 0.3% and 1.5% agar cultured for 12 h at 37 °C. (**B**) Analysis of swimming and swarming of WT, ∆*Vp-porin* and ∆*Vp-porin-C* strains in LB medium. The diameters of the swimming zone reflect bacterial migration on the 0.3% agar. The data are presented as the mean ± SD (n = 3). Columns have been marked with an asterisk (*** *p* < 0.001). (**C**) Visualization of bacterial flagella with negative-staining electron microscopy. Scale bar = 1 μm. (**D**) qRT-PCR analysis of the transcription levels of polar flagellar cluster I genes (*flgB*, *flgM*, *flgK*, *flgC*) and polar flagellar cluster II genes (fliE, fliK) in ∆*Vp-porin* compared to WT. The data are presented as the mean ± SD (n = 3). * *p* < 0.05; ** *p* < 0.01; *** *p* < 0.001. (**E**) qRT-PCR analysis of the transcription levels of lateral flagellar cluster I (*flgA*, *flgB*, *flgG*, *lafA*, *motY*) and lateral flagellar cluster II (*fliE*) genes in ∆*Vp-porin* compared to WT. The data are presented as the mean ± SD (n = 3). *** *p* < 0.001. Student’s *t* test was used to analyze ∆*Vp-porin* compared to WT. (**F**) qRT-PCR analysis of the expression levels of genes located in T6SS2 and *rpoN* in WT and ∆*Vp-porin* strains. The data are presented as the mean ± SD (n = 3). Student’s *t* test was used to analyze the different mutant strains compared to WT. * *p* < 0.05; *** *p* < 0.001.

**Figure 6 biology-13-00485-f006:**
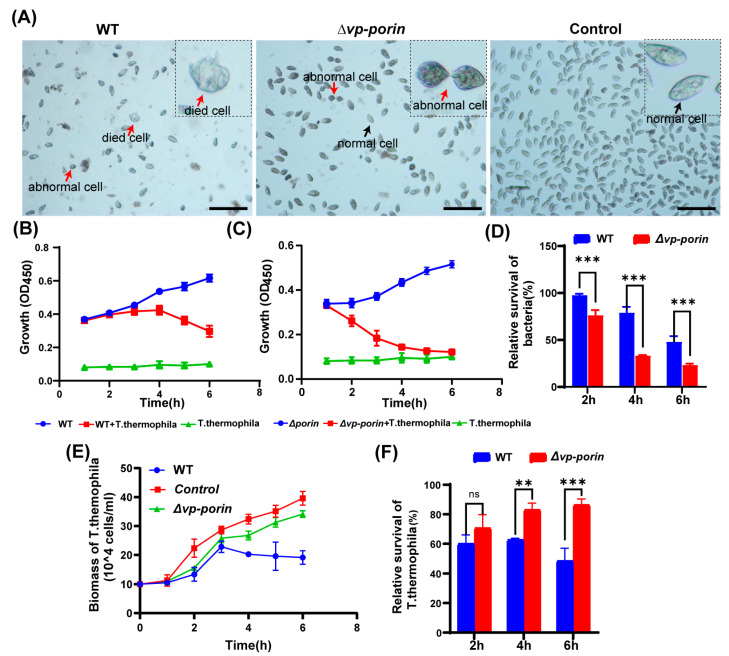
Assessment virulence of WT and ∆*Vp-porin* using *Tetrahymena*. (**A**) Morphological changes of *Tetrahymena* cells after co-cultivation with ∆*Vp-porin* and WT strains. Red arrows represent dead or abnormal cells. Black arrows represent live and normal cells. Scale bar: 200 μm. (**B**,**C**) Growth of ∆*Vp-porin* and WT co-cultured in the presence or absence of *Tetrahymena*. (**D**) Relative survival of ∆*Vp-porin* and WT strains co-cultured with *Tetrahymena*. The relative survival of bacteria was expressed as the OD450 value of strains co-cultured with *Tetrahymena* divided by that of bacteria grown alone at a different time. Data are expressed as the mean ± SD of three measurements per time point. *** *p* < 0.001. (**E**) Growth of *Tetrahymena* co-cultivated with ∆*Vp-porin* and WT strains. The control group was *Tetrahymena* grown alone in a sterile SPP medium. Data are expressed as the mean ± SD of three measurements per time point. (**F**) Relative survival of *Tetrahymena* co-cultivated with ∆*Vp-porin* and WT strains at 2 h, 4 h and 6 h. Relative survival of *Tetrahymena* is calculated by the number of *Tetrahymena* cells in the co-culture with different strains relative to that of *Tetrahymena* cells cultured alone. ** *p* < 0.01; *** *p* < 0.001; ns, not significant.

**Table 1 biology-13-00485-t001:** Strains and plasmids.

Strains	Genotype and Characteristics	Source
*V. parahaemolyticus 17802*	Cms, Kms, Ampr, wild-type strain,	ATCC
∆*Vp-porin*	*V. parahaemolyticus* strain in-frame deletion in *Vp-porin*	This study
∆*Vp-porin-C*	The complement of ∆*Vp-porin*	This study
*Escherichia coli*		
CC118	λpir lysogen of CC118 (Δ(ara-leu) araD ΔlacX74galEgalKphoA20 thi-1rpsE rpoB argE (Am) recA1	Our lab
CC118/pHelper	CC118 λpir harboring plasmid pHelper	Our lab
Plasmids		
pSR47S	Bacterial allelic exchange vector with sacB, KanR	Our lab
pSR47S-∆*Vp-porin*	A 1689 bp fragment containing the upstream and downstream sequences of the ∆*Vp-porin* gene in pSR47S, KanR	This study
pSR47S-*Vp-porin-C*	A 2634 bp fragment containing the *Vp-porin* sequences in pSR47S, KanR	This study

**Table 2 biology-13-00485-t002:** Sequences of PCR oligonucleotide primers.

Primer Name	Primer Sequence (5′ to 3′)	Purpose
UP-F	*CGAGCTCC*TTGATGGACTTCGCCAAC	Creation of ∆*Vp-porin* deletion fusion fragment
UP-R	CAACATTCGGTACTCAAGCAGCACTTGGTGCACGTTACTAC	
DOWN-F	GTAGTAACGTGCACCAAGTGCTGCTTGAGTACCGAATGTTG	
DOWN-R	*GACTAGT*GTACACACCGAATGCAGAC	
*Vp-porin*-T1	GAACAACACTAGAACGCGC	Confirmation of ∆*Vp-porin* deletion
*Vp-porin*-T2	TCGGTTACCGAAGAGTCTTC	

Note: Restriction sites are in italics. Complementary sites are underlined.

**Table 3 biology-13-00485-t003:** Primers used for q RT-PCR.

Primer Name	Primer Sequence (5′ to 3′)	Target
flgB-F	ACAAGGCACTAGGCATCC	polar flagellar cluster I genes
flgB-R	GACCATCTGTTCGGCTAAG	
flgC-F	GCGTCATGCTGTATTTGGTG	
flgC-R	AACCTGCACATTCGTTTGGT	
flgM-F	ATTCAAGTGCGACATCAAG	
flgM-R	CGGAGAAGCTGCCATATC	
flgK-F	GCCGTCAGTCAGTGATTC	
flgK-R	GTAGAGGACAGGTTGAGTTC	
fliE-F	CACTGTGCCCGTTTGCTTAC	polar flagellar cluster II genes
fliE-R	TCCGGCGGATGCTTCTATTC	
fliK-F	GTCGAGAAGAATGGCGAGAG	
fliK-R	CCAACTGAGCCTCTGACTCC	
flgA-F	TACCGACTGGCAAAGGTTGG	
flgA-R	TACCGACTGGCAAAGGTTGG	lateral flagellar cluster I genes
flgB-F	GCAGGTTCAGGCCCAGTATT	
flgB-R	TCATGTTGAGAAACGTCAGGCT	
flgG-F	AGATCTAGCGGTAATGGGGC	
flgG-R	GAGAAAGAGGTCGCGTTGTC	
lafA-F	GCTGGTGGCCTTATCGAAGA	
lafA-R	TACTGCGAAGTCTGCATCCAT	
motY-F	ATTAGTGAGGGTGCGCCTTT	
motY-R	GGTGAAGGGAAGGAATGGCA	
fliE-F	CGCTTGAGAAAACGACAGTGG	lateral flagellar cluster I genes
fliE-R	CCTACTAATGCGGTCTCGGC	
VPA1043-F	TCGAACAGCACGTAGAATCG	T6SS2 genes
VPA1043-R	GTGGCACTTCAGTTTCGTGA	
VPA1044-F	TCCTCAACCAAATCCTCGAC	
VPA1044-R	GCGTAGTTAGGCGTGTAGCC	
VPA1045-F	CCGATGCTCAATGGCTTAAT	
VPA1045-R	GCTGCTCTTTACCCAACTGC	
rpoN-F	GAGTGCACGGATTGCTGTTG	*rpoN* gene
rpoN-R	CGGTGGACATGCATGAATCC	
16s rRNA-F	TTAAGTAGACCGCCTGGGGA	qPCR of 16s rRNA
16s rRNA-R	GCAGCACCTGTCTCAGAGTT	

## Data Availability

The data that support the findings of this study are available from the corresponding author upon reasonable request.

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
