# Peer review of "The Impact of Vp-Porin, an Outer Membrane Protein, on the Biological Characteristics and Virulence of Vibrio Parahaemolyticus"

_biology, 2024, doi:10.3390/biology13070485_

Round 1

Reviewer 1 Report

Comments and Suggestions for Authors

The authors constructed a deletion mutant of Vp-porin, one of the outer membrane proteins of Vibrio parahaemolyticus, and analyzed its function. Vp-porin was involved in antibiotics permeability, and the deletion mutant strain showed reduced motility. Further functional analysis of Vp-porin in V. parahaemolyticus would be of significant interest.

In Figure 1, the position of the beta strand shown in (A) did not appear to match the position of the beta strand in (B). For example, beta strand 4 looked like #58-69aa in (A) and 70- next to alpha helix in (B). Please reconfirm them all.

The NCBI annotation “AMG08901.1” has the following description. What do you think about the difference from the author's 20-strand?

/note="Porins form aqueous channels for the diffusion of

small hydrophillic molecules across the outer membrane.

Individual 16-strand anti-parallel beta-barrels form a

central pore, and trimerizes thru mainly hydrophobic

interactions at the interface. Trimers...; cd00342"

Did the construction of the deletion mutant have any effect on the expression of the following genes adjacent to “AMG08901.1”?

"AMG08902.1"

 /region_name="OmpC"

/note="Outer membrane porin OmpC/OmpF/PhoE [Cell

wall/membrane/envelope biogenesis]; COG3203"

"AMG08903.2"

/region_name="RhaT"

/note="Permease of the drug/metabolite transporter (DMT)

superfamily [Carbohydrate transport and metabolism, Amino

acid transport and metabolism, General function prediction

only]; COG0697"

All experiments were only comparing the wild-type strain (parental strain) to the deletion mutant strain, but please do a restoration study on the complementary strain of Vp-porin.

In Figure 2E, F, compare the ratio of the diameter of the colony to the diameter of the plaque. Since the deletion mutants are slower growing, they cannot be compared by outer diameter due to differences in growth at the same 37°C, 24 hours.

In Figure 3, was the increased membrane permeability of the deletion mutant strain exhibited in changes in the morphology of the strain (e.g., electron microscopy)?

In Figure 5, you mentioned that both polar and lateral flagella showed decreased gene expression and reduced motility and swarming, but did you observe flagellar staining or the motility under optical microscopy? Has the generation of flagella been confirmed by electron microscopy? And also, please discuss what mechanism Vp-porin regulates flagellar gene expression.

Tetrahymena is a predator of bacteria and the following paper argues that the exotoxin produced by the bacteria is a defense against predators. On the other hand, the paper cited by the authors is aimed at evaluating the pathogenicity of Aeromonas to fish at low temperatures, such as in environmental water, and is not considered comparable to that of Aeromonas in humans.

Lainhart W. et al., Shiga Toxin as a Bacterial Defense against a Eukaryotic Predator, Tetrahymena thermophila. J Bacteriol. 2009 Aug; 191(16): 5116–5122. doi: 10.1128/JB.00508-09

Comments on the Quality of English Language

Please ask a native English speaker to correct your English.

Reviewer 2 Report

Comments and Suggestions for Authors

This was a really nice paper with clearly thought out experiments and a clear story. It was a pleasure to read and the conclusions in my opinion were not overstated and supported by the results. I can see there is a lot of mechanistic work to be done outside of the scope of this study and it will be interesting to see how this develops further. I don't have any major comments just a few small things:

1. Figure 2E: the images don't look like they support the text. It may just be this reviewer's eyes but do you have clear pictures?

2. Figure 2F: could you produce another figure correlating this data? It would be interesting to see if the decrease in size was matched between WT and mutant across the individual experiments.

3. Figure 6A: for the arrow I think you mean dead and not died cell. 

4. Figure 6A: could you crop around some of the key phenotypes for the reader? 

5. Figure 6A: as the phenotypes are really interesting-do you have a way to quantify your images automatically to get a sense of the number of each e.g., number of normal, vs dead ? 

Overall, a great piece of work with a lot of appeal. Well done to the authors. 

Round 2

Reviewer 1 Report

Comments and Suggestions for Authors

I agree with you for Responces 1 and 2.

Comments 3:

Did the construction of the deletion mutant have any effect on the expression of the following genes adjacent to “AMG08901.1”?

"AMG08902.1" /region_name="OmpC"/note="Outer membrane porin OmpC/OmpF/PhoE [Cell wall/membrane/envelope biogenesis]; COG3203" 

"AMG08903.2"/region_name="RhaT"/note="Permease of the drug/metabolite transporter (DMT) superfamily [Carbohydrate transport and metabolism, Amino acid transport and metabolism, General function prediction only]; COG0697"

Response 3:

Thanks for your comments. We have confirmed the mRNA expression level of the two genes using qRT-PCR analysis. The results indicated the mRNA expression levels of both ompC and rhotL genes were downregulate in ∆Vp-porin.

For Response 3, was the ompC promoter in the Vp-porin gene or was the Vp-porin direct/indirect regulating the expression of the ompC gene? In either case, it is suggested that all of the experiments in this manuscript may not be a direct effect of the Vp-porin gene deletion. That is, the effect of Vp-porin on substance transport needs to be properly reevaluated.

I agree with you for Responce 4. In relation to this, you were using the overlap PCR method to construct the deletion mutant, but the primers UP-R and DOWN-F hybridized perfectly as shown below, but I think you should provide details on how to suppress this hybridization to perform PCR.

UP-F   5' CAACATTCGGTACTCAAGCAGCACTTGGTGCACGTTACTAC 3'

DOWN-R 3' GTTGTAAGCCATGAGTTCGTCGTGAACCACGTGCAATGATG 5'

I agree with you for Responces 6 and 7.

For your Responce 8, again, since Tetrahymena is a predator of bacteria, isn't a factor on the bacterial side to resist it a virulence factor? If authors mean in relation to phagocytosis by macrophages, shouldn't you clarify the difference between them? Also, please mention this in the Discussion.
